evolution, palaeontology

crocodile, skull, evolutionary rate, convergent evolution, three-dimensional morphometrics

**Author for correspondence:**
Ryan N. Felice
e-mail: ryan.felice@ucl.ac.uk

# Complex macroevolutionary dynamics underly the evolution of the crocodyliform skull

Ryan N. Felice[1,2], Diego Pol[3] and Anjali Goswami[2]

[1]Centre for Integrative Anatomy, Department of Cell and Developmental Biology, University College London, London, UK
[2]Department of Life Sciences, The Natural History Museum, London, UK
[3]CONICET, Museo Paleontológico Egidio Feruglio, Trelew 9100, Chubut, Argentina

RNF, 0000-0002-9201-9213; DP, 0000-0002-9690-7517; AG, 0000-0001-9465-810X

All modern crocodyliforms (alligators, crocodiles and the gharial) are semi-aquatic generalist carnivores that are relatively similar in cranial form and function. However, this homogeneity represents just a fraction of the variation that once existed in the clade, which includes extinct herbivorous and marine forms with divergent skull structure and function. Here, we use high-dimensional three-dimensional geometric morphometrics to quantify whole-skull morphology across modern and fossil crocodyliforms to untangle the factors that shaped the macroevolutionary history and relatively low phenotypic variation of this clade through time. Evolutionary modelling demonstrates that the pace of crocodyliform cranial evolution is initially high, particularly in the extinct Notosuchia, but slows near the base of Neosuchia, with a late burst of rapid evolution in crown-group crocodiles. Surprisingly, modern crocodiles, especially Australian, southeast Asian, Indo-Pacific species, have high rates of evolution, despite exhibiting low variation. Thus, extant lineages are not in evolutionary stasis but rather have rapidly fluctuated within a limited region of morphospace, resulting in significant convergence. The structures related to jaw closing and bite force production (e.g. pterygoid flange and quadrate) are highly variable, reinforcing the importance of function in driving phenotypic variation. Together, these findings illustrate that the apparent conservativeness of crocodyliform skulls betrays unappreciated complexity in their macroevolutionary dynamics.

## 1. Introduction

Archosaurs are represented by two extant groups: birds, with over 10 000 living species, and crocodylians, with fewer than 30. These lineages diverged approximately 250 Ma, in the immediate aftermath of Permo-Triassic mass extinction, and both survived two further mass extinctions, 201 and 66 Ma [1,2]. Bird-lineage archosaurs evolved extensive ecological, morphological and taxonomic diversity in the Mesozoic and Cenozoic eras, and continue to display extensive diversity despite suffering extinction of all non-avian dinosaurs in the Cretaceous–Palaeogene mass extinction. By contrast, extant crocodyliforms are species poor, restricted ecologically to semi-aquatic habitats and faunivorous diets, and show limited diversity in phenotype, resulting in common dismissal of the clade as 'living fossils' [3]. This lack of extant variation, however, hides a richer extinct diversity [4,5]. Mesozoic forms occupied a much wider range of ecologies, including fully pelagic taxa and multiple independently evolved herbivorous and omnivorous forms [6,7]. This wider niche breadth was accompanied by greater variation in morphology, especially in regard to craniodental anatomy [4–8]. Nonetheless, crocodyliform cranial variation is small compared to that of dinosaurs (including birds), the only other archosaur clade to survive to the present day. While cranial phenotypes related

to edentulism, nectivory and other specialized trophic niches evolved multiple times in dinosaurs, none of these are known from Crocodyliformes. This striking divergence in evolutionary outcomes begs the question: what has constrained crocodyliform cranial form and function?

The crocodyliform skull shape is thought to vary primarily in the degree of snout (i.e. maxillary rostrum) elongation [4,5,8–12]. Length and width of the snout are correlated with response to biomechanical stress and hydrodynamic properties and are thought to evolve in response to maximum prey size and other aspects of trophic ecology [8,9,11]. However, the importance of snout elongation in crocodyliform evolution may be exaggerated as a result of insufficiently quantifying the variation in other regions of the skull. Nearly all studies of crocodyliform skull shape quantify morphology using two-dimensional two-dimensional geometric morphometrics, with landmarks placed almost entirely on the outline of the skull [4,5,8] or the outline and dorsal surface, usually from dorsal view photographs [6,9,13,14]. This approach excludes important information about the ventral and posterior surfaces of the skull, skull height, the shape and size of the adductor chamber, palate anatomy and the configuration of the individual elements contributing to gross skull morphology. Given that the vertebrate skull is a complex three-dimensional structure, quantifying its variation using such simple two-dimensional measurements may inadvertently obscure the detail and complexity of its evolutionary history in crocodyliforms, essentially driving a simple narrative by data choice.

The few analyses of three-dimensional skull shape in crocodyliforms to date have largely been phylogenetically restricted to extant clades [15,16]. However, these studies have demonstrated the importance of numerous features beyond anteroposterior length. For example, alligatoroids and crocodyloids differ in the positions of the pterygoid and the articular surface of the quadrate, suggesting that craniomandibular articulations are an important part of skull variation in extant forms [16]. Similarly, caimans show high variation in the pterygoid flange [15], which is a key structure in resisting biomechanical forces during feeding by acting as an 'open' buttress joint against the mandible [11,17–19]. Furthermore, the dorsal surface of the pterygoid bone serves as the origin for the pterygoideus anterior muscle [20,21], which is critical for producing different biting behaviours, including high bite forces in carnivorous taxa and masticatory movements in herbivorous and omnivorous taxa [20,21]. Innovations in palate morphology are also important aspects of crocodylomorph evolution but are excluded entirely from dorsal-only analyses. For example, different groups have independently acquired a derived secondary palate formed by the palatine and pterygoid bones, which in neosuchians is thought to be an adaptation for semi-aquatic life [22] and for resisting the torsional and compressive forces experienced during feeding [3,18].

Here, we conduct, to our knowledge, the first three-dimensional study of skull evolution spanning the breadth of crocodyliform diversity, using high-density three-dimensional geometric morphometrics (1291 three-dimensional landmarks and semilandmarks (electronic supplementary material, figure S1) to comprehensively quantify the skull across extant ($n = 24$) and extinct taxa ($n = 19$). High-density geometric morphometrics seeks to quantify shape data in extremely high detail by distributing landmarks and semilandmarks across the entire surface of the structure of interest, capturing much more nuanced information

about phenotypic variation than more simplistic morphometric techniques [23,24]. This approach has been applied to diverse vertebrate clades in recent years and enables the representation of the entire surface of the skull, capturing key structures that are typically overlooked [25–27]. With these data, we use phylogenetic comparative methods and evolutionary modelling to reconstruct the evolution of the crocodyliform skull over 200 Myr and test long-standing hypotheses on how convergence and constraint have shaped its evolution. First, we quantify variation across the skull across the whole of crocodyliform diversity and individually for each cranial element, providing a comprehensive three-dimensional perspective of the shape of this highly complex structure. We then investigate how cranial shape differs across lineages and test the influence of phylogeny, diet, convergent evolution and evolutionary allometry, using multiple phylogenetic topologies to control for uncertainty in phylogenetic relationships and divergence estimates. We further examine how the rate of morphological evolution has changed through time and across clades and assess whether the conservativeness of cranial form in extant crocodyliforms results from decelerated evolution or extensive convergence.

Finally, we interrogate how evolutionary patterns differ across the anatomical structures that make up the skull. The individual components of a complex phenotype may be highly correlated or 'integrated' with one another, in which case the entire system is expected to exhibit a coordinated evolutionary change in response to selection. Conversely, complex phenotypes may be parcelled into quasi-independent 'modules' that exhibit distinct evolutionary histories. The organization of traits into modules reflects developmental and functional correlations among these traits. By studying the semi-independent evolutionary histories of phenotypic modules we can gain insights into how trait interactions among traits have constrained or facilitated the evolution of disparity [28,29].

## 2. Results

### (a) Cranial variation and convergence

Principal components analysis (PCA) confirms the prevalence of homoplasy in crocodyliform cranial evolution (figure 1a). The first principal component axis (PC1) accounts for 55.9% of the total variance and describes the length to width ratio of the snout, skull height and the relative position of the quadrate condyles relative to the occipital condyle (figures 1 and 2). The marine thalattosuchians and the partially piscivorous *Gavialis* and *Tomistoma* score high on PC1, as do *Pholidosaurus* and *Mecistops*. These taxa are distantly related to each other (figure 3), representing at least three independent origins of this phenotype. The highly terrestrial notosuchians have low PC1 scores, with crocodylians exhibiting intermediate scores. The second principal component axis (PC2; 12.5% of total variance) is dominated by variation generally excluded in two-dimensional dorsal view studies, particularly the size of the pterygoid flange, the dorsoventral flexion of the dorsal surface of the snout, as well as the position of the eye (dorsally versus laterally oriented), most members of Alligatoroidea and Crocodyloidea score high on PC2 and have dorsal orbits, large pterygoid flanges and a dorsal concavity of the snout. All other clades are low on PC2, with more lateral orbits, smaller pterygoids and more dorsoventrally compressed snouts. PC3 (5.5% of the total variance) is dominated by the dorsoventral tapering of the snout,

Proc. R. Soc. B 288: 20210919

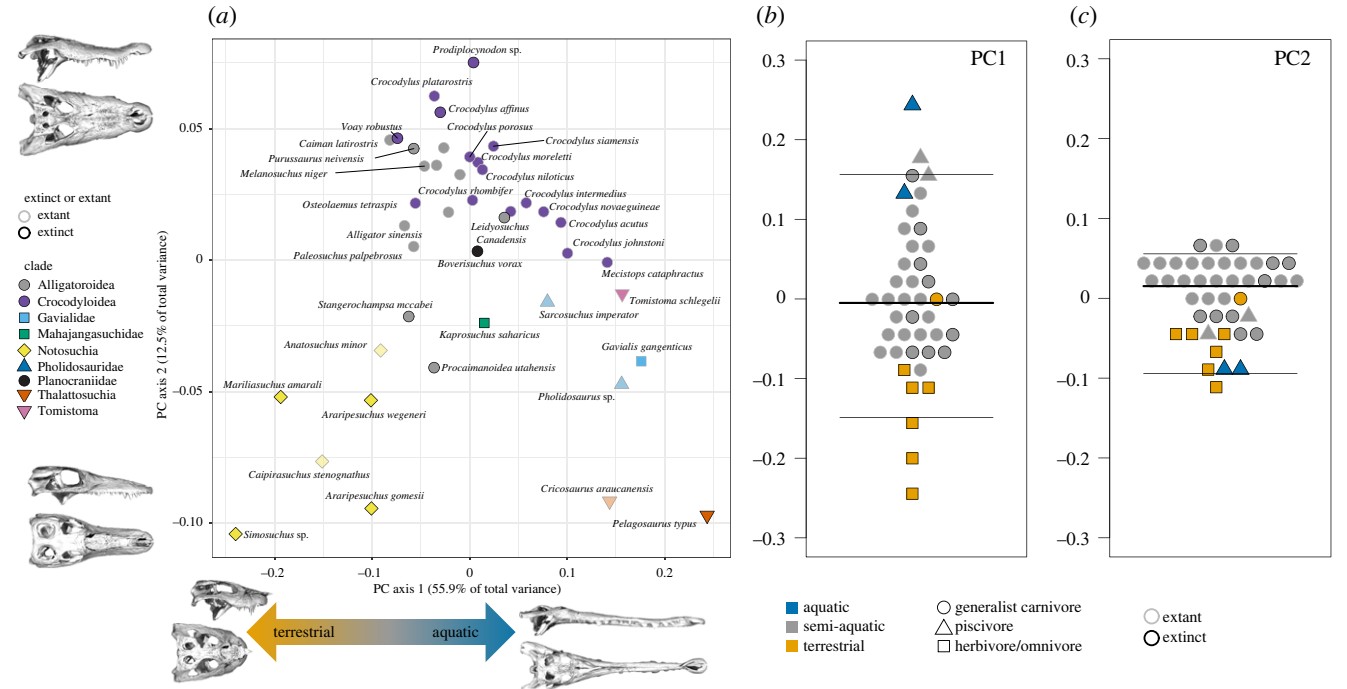

**Figure 1.** Principal component analysis. The first two principal components (*a*) represent a total of 68.4% of the cumulative variance. Specimens that were partially reconstructed are indicated with translucent points. Skull models indicate the shape change described by each axis. PC axis 1 separates taxa according to habitat and diet (*b*). PC axis 2 separates the extant alligators, caimans and crocodiles from other groups (*c*). (Online version in colour.)

whereas PC4 (4.2% of the total variance) reflects its lateral tapering (electronic supplementary material, figure S2).

Whereas PCs 1 and 4 capture aspects of variation that are represented in part in two-dimensional studies using dorsal view, the variation captured by PCs 2 and 3 is almost entirely excluded from that view. Exploded skull views of the variation of individual cranial modules on the positive and negative extremes of PCs 1 and 2 further show the complexity of crocodyliform cranial variation (figure 2). Alongside changes in skull length are substantial shifts in skull height, the orientation and size of processes such as the pterygoid flange, the postorbital bar, and jaw joint, and the curvature of the entire snout, and dorsoventral expansion of individual elements such as the maxilla, premaxilla, quadratojugal.

PC1 has a strong ecological component, separating terrestrial herbivores and omnivores from piscivores (figure 1*b*). Quantifying phenotypic vectors confirms that there is significant convergent evolution in longirostrine species, with the angle of multivariate evolution among these taxa smaller than expected by chance ($\theta = 35.8$, $p = 0.001$) [30]. As expected, however, incorporating a more comprehensive three-dimensional quantification of skull morphology complicates its association with ecology: the same terrestrial herbivorous/omnivorous taxa and marine piscivorous taxa that define opposite extremes of PC1 converge with low scores on PC2 owing to a shared small pterygoid flange (figure 1*c*). Phylogenetic MANOVA further demonstrates that the relationship between diet and skull morphology is contingent on phylogenetic relationships. There is not a significant relationship between diet and skull shape independent of phylogeny under any of the three phylogenetic hypotheses dated with fossilized birth–death tip dating ($p = 0.63$–$0.85$). However, using the topology with node dates derived from molecular clock dating (tree 4) recovers a weak but significant relationship between diet and skull shape ($R^2 = 0.11$, $p = 0.01$).

Because extinct crocodyliforms occupied a greater breadth of spatial and trophic niches than extant forms, it is expected that extant species also display lower cranial disparity than stem clades. PCA confirms that Alligatoroidea and Crocodyloidea indeed occupy a limited region of morphospace (figure 1*a*), and extinct taxa represent approximately three times greater Procrustes variance than extant taxa (0.024 versus 0.008, $p = 0.001$).

Allometry and phylogenetic history are often considered key factors in shaping and constraining crocodyliform variation and diversification [4,16,31,32]. Allometry has a significant but relatively small effect on overall skull shape (electronic supplementary material, table S1; $R^2 = 0.21$, $p = 0.001$), similar to other vertebrate clades [25–27,33]. Phylogenetically informed regressions of skull shape on log-centroid size across the four phylogenetic hypotheses recover a similar significant but weak relationship ($R^2 = 0.07$–$0.21$, $p = 0.002$–$0.02$). Multivariate phylogenetic signal [34] is also significant but weak across phylogenetic hypotheses dated using the fossilized birth–death model (trees 2–4, $K_{mult} = 0.04$–$0.10$, $p = 0.001$), but is higher under a phylogenetic hypothesis with node dates derived from molecular clock estimates (tree 1, $K_{mult} = 0.47$, $p = 0.001$).

## (b) Tempo of crocodyliform cranial evolution

Bayesian evolutionary modelling demonstrates that the evolutionary history of the crocodyliform skull is best described by a variable-rates Brownian motion model, rather than a single rate Brownian motion or Ornstein–Uhlenbeck (OU) model (Bayes factor greater than 10; electronic supplementary material, figure S2). Rates of phenotypic evolution are heterogeneous through time and across lineages (figure 3; electronic supplementary material, figure S3). There are particularly high rates of evolution in modern Crocodylidae across all four

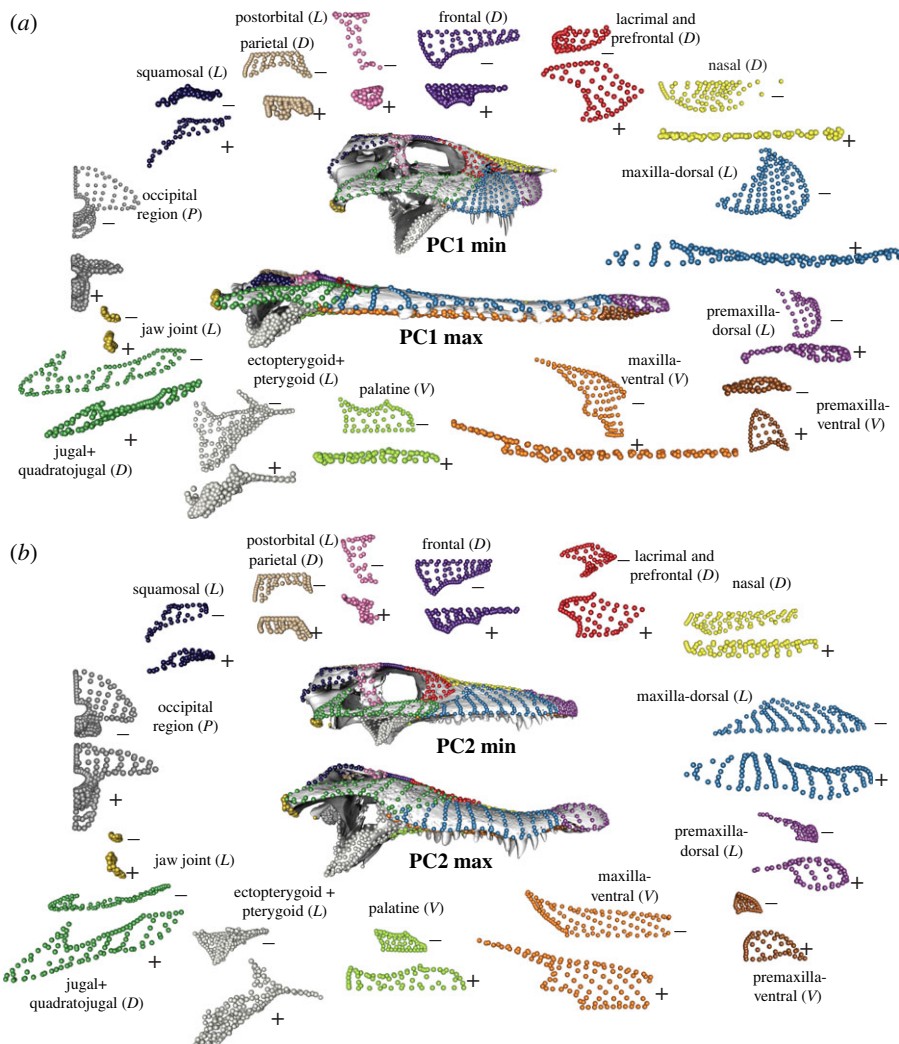

**Figure 2.** Shape change along principal component (PC) axes 1 (*a*) and two (*b*). Skull models represent the hypothetical shapes at the extreme of each PC axis and were generated by warping a model of the skull of *Alligator mississippiensis* to fit the landmark configuration representing the shape at each extreme. The coloured landmark configurations around each skull represent the shape change in individual modules along that PC axis, with the shape at the minimum and maximum shapes marked with − and +, respectively, and skull orientation indicated as (*D*) dorsal, (*L*) lateral, (*P*) posterior or (*V*) ventral. (Online version in colour.)

phylogenetic hypotheses. Using a phylogenetic hypothesis with node dates derived from molecular clock estimates, we recover a high posterior probability of a shift in rates at the most recent common ancestor of *Crocodylus johnstoni* and *Crocodylus siamensis* (figure 3 and electronic supplementary material, figure S3A). This is the only region of the tree that is consistently recovered as having a high probability of rate shift (posterior probability greater than 0.70) in all four phylogenies (electronic supplementary material, figure S4). The exact position of this rate shift differs across the four phylogenetic hypotheses, variably including *C. johnstoni*, *Crocodylus mindorensis*, *Crocodylus novaeguineae*, *Crocodylus palustris*, *Crocodylus porosus* and *C. siamensis*. Regardless of the position of this rate shift on the tree, these rapidly evolving species represent a clade distributed throughout southern Australia, southeast Asia and the Indo-Pacific.

The morphologically diverse Notosuchia is estimated to have sustained high rates of evolution under two out of the four phylogenetic hypotheses (figure 3; electronic supplementary material, figure S4A,B) and Thalattosuchia in just one (figure 3). None of the nodes in Notosuchia or Thalattosuchia have high posterior probabilities of a rate shift in any scenario, but there is a high probability shift towards slower evolutionary rates near the base of Neosuchia in every topology. While some higher rates of evolution are

observed in branches leading to Cretaceous and Palaeogene neosuchians, this is highly dependent on topology and high rates of evolution in Neosuchia that are supported across all topologies are not observed until the Neogene.

## (c) Modularity, integration and mosaic evolution of the crocodyliform cranium

We quantified the levels of integration, disparity and evolutionary rate across the individual cranial elements (i.e. morphological modules) that make up the skull, demonstrating that the crocodyliform skull exhibits mosaic evolution, with disparity and evolutionary rate varying across anatomical modules. The modules with the highest disparity (Procrustes variance) are those that contribute to snout shape (maxilla, premaxilla), orbit shape and position (jugal, lacrimal, prefrontal) and craniomandibular joints (pterygoid and 'jaw joint', i.e. articular surface of the quadrate; electronic supplementary material, figure S5). Modules with low disparity include elements of the braincase (occipital region, frontal, parietal) and the postorbital. There is no consistent correlation between the phenotypic disparity (Procrustes variance) and the magnitude of within-module evolutionary integration (electronic supplementary material, figure S5). The jaw joint has the highest

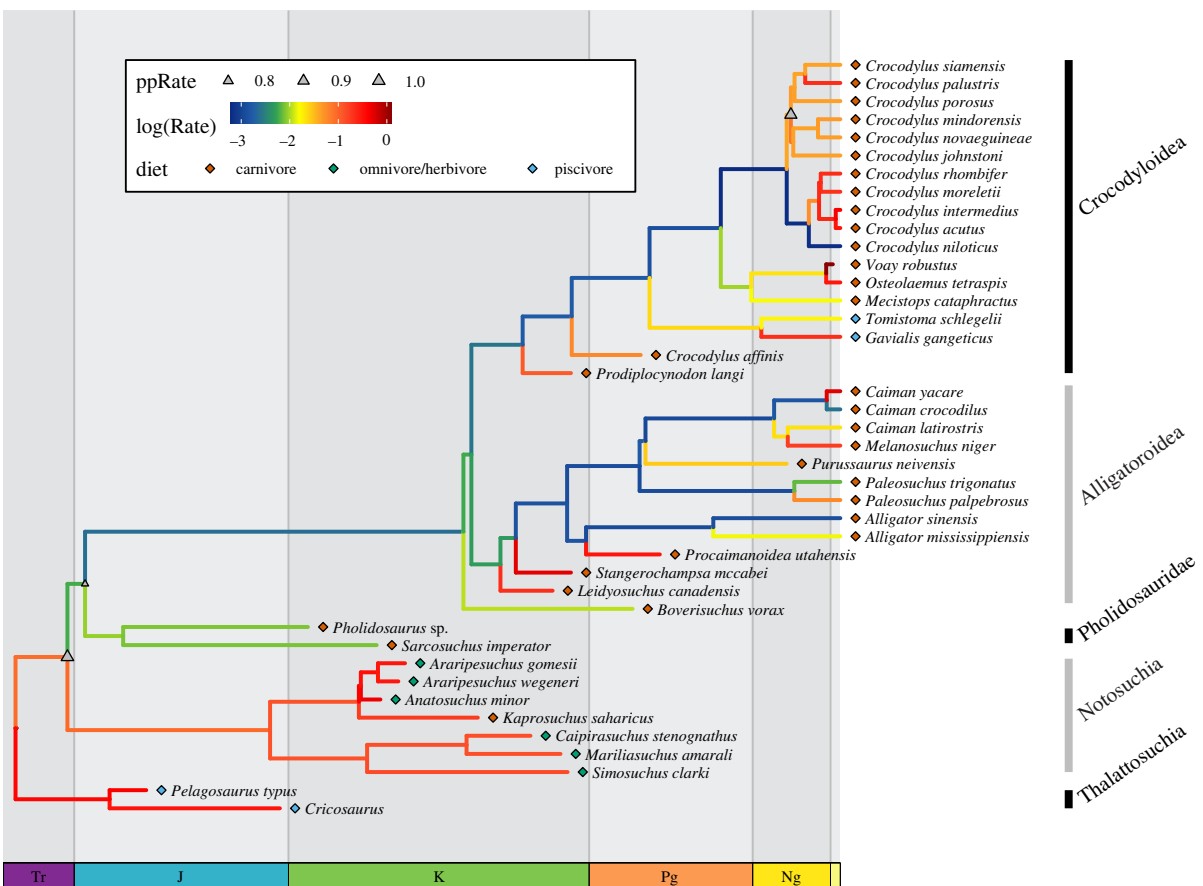

**Figure 3.** Variable-rates Brownian motion model of skull evolution. Time-scaled phylogeny with branches coloured according to log-transformed mean rate scalar from the posterior distribution recovered from BayesTraits. Coloured diamonds indicated the diet of the taxon. Triangles indicate the posterior probability of a clade-wise rate shift (ppRate) on the corresponding node, with the size of the triangle proportionate to the probability of the shift. (Online version in colour.)

within-module integration but moderate disparity. Modules with moderate within-module integration values span from very low disparity (occiput, parietal, postorbital) to very high disparity (premaxilla). Evolutionary integration was calculated from the phylogenetic independent contrasts of shape, and different phylogenetic hypotheses result in minor differences in integration values. In particular, phylogenetic hypothesis 3 shows a somewhat lower integration in the parietal and jugal/quadratojugal modules and higher integration in the frontal and nasal modules. Nonetheless, overall integration patterns are largely consistent across phylogenetic hypotheses.

Examining the Procrustes variance and evolutionary rates of the individual landmarks demonstrates that these are diffusely distributed across the crocodyliform skull (figure 4; electronic supplementary material, figure S6), rather than the more concentrated patterns observed in other tetrapod clades [25,27,28]. Many skull regions experience high variability, including the pterygoid, ectopterygoid, jugal, quadratojugal and squamosal. Rate and disparity are heterogenous even within individual cranial bones. Within the pterygoid, there are high rates of evolution in the lateral end of the pterygoid flange and low rates around the choana. Similarly, the lateral portion of the maxilla has higher rates and disparity than the medial portion.

## 3. Discussion

Crocodiles have long been described as living fossils, with conserved cranial morphology that varies in only one aspect: snout

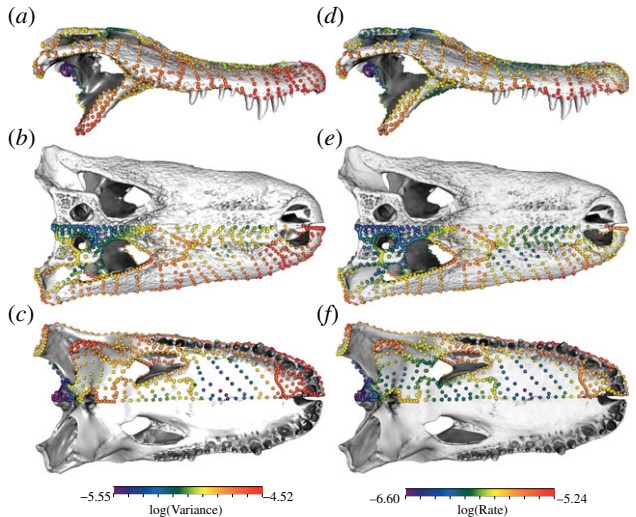

**Figure 4.** Per-landmark disparity and evolutionary rate. Disparity is calculated as Procrustes variance and illustrated in lateral (*a*), dorsal (*b*) and ventral (*c*) view. Rate is calculated using the multivariate implementation of Blomberg's *K*, using tree 1. Illustrated here in lateral (*d*), dorsal (*e*) and ventral (*f*) view. (Online version in colour.)

elongation. This characterization belies a rich extinct diversity, with fossil crocodyliforms filling a much broader range of ecological niches than observed in extant forms. This simplistic portrayal of crocodyliform evolution is partly driven by the data used to quantify their morphology. Whereas three-dimensional morphometrics has long been the standard for

comparative analyses of vertebrate cranial evolution [35–39], studies of crocodyliforms still rely overwhelmingly on two-dimensional geometric morphometric data. Even studies that celebrate their extinct diversity generally measure it using dorsal view images that exclude much of the complexity of the cranium. Here, by using a high-density three-dimensional morphometric approach capable of comprehensively capturing the morphology of the entire cranium, as well as individual elements, we demonstrate that snout elongation is a major, but certainly not the only trait contributing to patterns of cranial variation across crocodyliforms. Substantial variation in observed in every cranial element (figures 2 and 4) and the high disparity is distributed across the crocodyliform skull, particularly in the pterygoid, orbit and quadratojugal, in addition to the snout (figure 4). This pattern is unusual among extant tetrapods, with the other extant archosaur lineage, birds, showing an overwhelming concentration of cranial variation in the anterior rostrum [26,28], while squamates and amphibians show concentrations of high disparity in the elements forming the suspensorium [27,33,40]. A more diffuse pattern of cranial disparity and evolutionary rates also characterizes non-avian dinosaurs [25], which may reflect a greater diversity of trophic niches and food acquisition strategies.

Within the snout, our results support the hypothesis that convergent evolution of an elongate snout is a key component of crocodyliform cranial evolution [8,10], with multiple independent origins of this morphotype. The brevirostrine–longirostrine axis has a strong ecological signal, forming a gradient from short-faced terrestrial omnivorous/herbivorous taxa to long-faced marine taxa with generalist carnivores in between (figure 1). Although we do not recover a significant difference in skull shape between dietary groups under most phylogenetic hypotheses, this is probably owing to (i) diet being highly correlated with phylogeny, with a single origin of omnivory/herbivory in the current sample, and (ii) relatively few members of the herbivore/omnivore and piscivore guilds compared to the generalist carnivore guild. Nonetheless, there is significant convergent evolution in long-snouted forms, such as *Gavialis*, *Tomistoma*, *Mecistops*, *Pholidosaurus*, *Pelagosaurus* and *Cricosaurus*, representing at least three independent lineages, supporting the hypothesis that cranial function is a key driver of skull shape evolution [4,9,11].

Importantly, our approach identified that variation in the snout is more complex than simple elongation. The cross-sectional shape of the snout is also a key feature separating aquatic and piscivorous taxa from terrestrial and omnivorous/herbivorous taxa. Snout cross-sectional morphology is linked with diet and biomechanical performance, with platyrostral taxa (i.e. those with wide and dorsoventrally compressed snouts) being more resistant to mediolateral bending and oreinirostral taxa (i.e. those with high and mediolaterally compressed snouts) being more resistant to dorsoventral bending [9,10,18]. Mediolateral and dorsoventral compression of the snout are partially decoupled—PC1 describes both snout length and mediolateral width, while PC2 describes the lateral profile of the snout including dorsoventral compression. Dorsoventral compression is a key aspect of variation among brevirostrine crocodylians (e.g. alligatoroids and crocodylids other than Gavialoidea) and separates brevirostrine crocodylians from other lineages (figure 1*c*). PC3 and PC4 also reflect dorsoventral and lateral tapering of the snout, respectively,

highlighting the importance of capturing skull shape in three dimensions.

Beyond the snout, extensive variation is observed in the cranial elements that form joints with the mandible: the quadrate and the pterygoid flange (figures 2 and 4). The pterygoid is closely linked with feeding performance as it resists forces during biting and serves as the attachment point for the pterygoideus muscles [11,17]. Notosuchian herbivores and omnivores were capable of chewing-like fore-aft motions of the mandible using the pterygoideus muscles and the configuration of their pterygoid bone and jaw articulations [21,41]. Extant taxa, especially Alligatoroidea and Crocodylidae, occupy a restricted region of morphospace characterized by a large pterygoid flange and posteriorly positioned quadrate. These traits are potentially correlated with producing high bite force, a functional trait which is hypothesized to have contributed to the evolutionary longevity and success of neosuchians as semi-aquatic predators [42]. Crocodylids and alligatoroids both have high variation along PC2 and thus in the morphology of these functional traits (figure 1), supporting the conclusion that even extant crocodyliforms have largely unappreciated diversity in cranial form and function [8].

The surprising variation in extant crocodyliforms is further supported by analyes of evolutionary rates. Only the extant Crocodyloidea exhibits a high probability shift to sustained high rates of evolution across all phylogenetic hypotheses. Faster rates in crocodiles than in alligators is consistent with previous studies which found that alligators have higher phylogenetic inertia than crocodiles [16]. More fundamentally, this result suggests that, despite having low overall disparity, modern crocodyloids are not experiencing evolutionary stasis. Instead, they are rapidly and repeatedly exploring a limited range of phenotypes. All phylogenetic hypotheses recovered a high posterior probability of a rate shift within Crocodyloidea. The position of this shift varies slightly across the four trees, but always comprises a clade of taxa that are found in Australia, southeast Asia and the Indo-Pacific. The high rate of phenotypic evolution in this geographically restricted clade reinforces the importance of abiotic environmental factors in driving phenotypic evolution.

Outside of this recent diversification of crocodyloids, neosuchians are consistently outpaced by their extinct relatives. Notosuchians and thalattosuchians shower higher rates of evolution than is observed in branches leading to Neosuchia or within most Cretaceous and Palaeogene Neosuchian lineages. Although there are no high probability rate shifts within these extinct crocodyliform clades, there is consistently a downshift in evolutionary rates near the base of Neosuchia. Coupled with the demonstration that extant taxa display approximately one-third of the disparity of all crocodyliforms, it is evident that Cenozoic crocodyliforms are poor representatives of their larger clade in terms of shape, variation or evolutionary tempo. In that sense, the term living fossils as applied to modern crocodyliforms is inappropriate in multiple senses. First, Neogene crocodyliforms are unusually fast evolving for the clade, meaning that they are not in stasis, but rapidly shifting within a relatively narrow morphological range. Second, the term suggests that the clade as a whole shows relatively little change, whereas the extinct Mesozoic crocodyliforms were not only more variable in shape and ecology but were also faster evolving than most of the crown group. Were extant crocodyliforms actually similar to the early members of their clade,

they would be considerably more diverse and faster evolving that what we observe in the modern fauna.

Why did evolutionary rates slow down at or near the base of Neosuchia, and why do crown crocodyliforms, even when fast evolving, occupy a relatively limited area of cranial morphospace compared to their extinct relatives? Explanations for these patterns may be derived from extrinsic or intrinsic factors, or both. Competition with mammals in the terrestrial and aquatic realms in the Cenozoic may have suppressed crocodyliforms from exploiting the same niches as their extinct relatives and limited them to the semi-aquatic niche that all living forms occupy [12]. Specialization to a specific habitat or niche itself can also create an evolutionary ratchet, limiting the ability to diversify into other niches, as is observed in mammalian hypercarnivores [43]. Extant crocodyliforms are, seemingly in contradiction to the hypothesis of an evolutionary ratchet, semi-aquatic generalists. However, crocodyliforms display numerous adaptations to their semi-aquatic niche, from the structure of the vertebral column and limbs [19,44]. The requirement to be functional in both terrestrial and aquatic environments may disadvantage taxa in competition with specialists in either of those environments and thus limit their capacity to reinvade either niche. It may also result in antagonistic selection between more terrestrial and more aquatic specializations, limiting the ability to evolve in either direction. That requirement to maintain functionality in both niches may thus constrain and slow their evolution, as has been observed in other semi-aquatic vertebrates such as frogs [40].

Beyond extrinsic factors, intrinsic factors may limit variation and, ultimately, evolution. Evolutionary integration of traits can arise from many processes, including genetic correlations among traits to co-selection of traits, both of which may affect the disparity of those traits across species. Other archosaur clades display a strong negative correlation between the magnitude of integration and disparity within modules, with highly integrated modules displaying low disparity, suggesting that the processes which drive the correlated evolution of traits may also limit their variation [28,29]. By contrast, crocodyliforms exhibit a weak relationship between phenotypic disparity and evolutionary integration (electronic supplementary material, figure S5). Interestingly, within-module integration and disparity of the neurocranium (e.g. occipital, postorbital) are low in both non-avian dinosaurs and crocodyliforms [25], but high in birds [26], suggesting that this is a derived condition in birds potentially related to changes in development and cranial architecture and function [45].

Applying a three-dimensional approach demonstrates the macroevolutionary history of the crocodyliform skull is much more dynamic than previously appreciated. The interactions among ecology, allometry and phylogeny have generated a wide range of skull morphologies and substantial convergences within this clade of its 230 Myr history. Phenotypic variation in crocodyliforms is not restricted to changes in snout length but involves an unusually diffuse pattern of high disparity across cranial regions related to food acquisition and jaw mechanics. Although extant crocodyliforms indeed exhibit much less phenotypic variation than their extinct relatives, and as a whole have downshifted in evolutionary rate near the base of Neosuchia, their evolution is far from static. Indeed, modern crocodiles are among the fastest evolving species in the clade. Nonetheless, their evolution

seems to be limited to a relatively small region of cranial morphospace, but whether owing to extrinsic or intrinsic factors requires further investigation. Together, these new insights into cranial evolution in Crocodyliformes reinforce the necessity of moving past simplistic quantifications of phenotypes towards high-dimensional, phenome-scale approaches to reveal cryptic complexity in the evolution of organismal diversity.

# 4. Methods

## (a) Sampling
We quantified cranial morphology in a sample of 42 fossil and modern crocodylomorph species spanning the phylogenetic breadth of the clade (electronic supplementary material, table S2). Specimens represent wild-caught individuals estimated to be adults on the basis of body size and cranial morphology. Sex is unknown for most specimens. Three-dimensional models of most specimens were generated via surface scanning with Creaform GO!Scan20 and FARO ScanArm scanners with additional specimens obtained from computed tomography scans and online databases (electronic supplementary material, table S2). We only included fossil specimens that were preserved in three dimensions, complete and relatively undeformed. Five specimens lacked ventral elements (e.g. pterygoid, palatine) owing to incomplete preservation or preparation. Landmarks for the unpreserved regions of these five specimens were estimated with thin-plate splines using the function 'fixLMtps' in the 'Morpho' R package [46]. For each species, ecological traits were scored based on published ecological surveys and functional analyses of fossils [7,8].

## (b) Three-dimensional geometric morphometrics
We quantified the external morphology of the skull using a semi-automated three-dimensional surface landmarking approach [46,47]. Anatomical landmarks (102 three-dimensional points) and semilandmark curves (665 three-dimensional points) were placed on the digital skull models using Stratovan Checkpoint (electronic supplementary material, table S3) and were selected in order to isolate 15 cranial regions. Regions are displayed in detail in the electronic supplementary material, figure S1, with landmarks and curves detailed in the electronic supplementary material, table S4. We then digitized the same landmarks and semilandmarks on a template model and added to this 597 surface semilandmarks. This template was used as an atlas to automatically project surface semilandmarks onto the specimens using the 'Morpho' R package [46], resulting in 1364 total three-dimensional landmarks and semilandmarks. Landmarks were digitized bilaterally and semilandmarks were only digitized on the right side of the skull. Right side landmarks were reflected across the midline to create a bilaterally symmetrical configuration to minimize alignment artefacts [48]. These mirrored landmarks were then removed after Procrusted alignment and before subsequent analyses.

## (c) Phylogenetic hypotheses
The phylogenetic relationships within Crocodylomorpha are debated, with the relationship between gavialids and *Tomistoma* and the position of Thalattosuchia with respect to Neosuchia as key uncertainties. As such, we used three different phylogenetic topologies—(A) *Gavialis* outside of 'Brevirostres' and thalattosuchians as neosuchians, (B) *Gavialis* outside of 'Brevirostres' and thalattosuchians as non-neosuchians, and (C) *Gavialis* and *Tomistoma* as sister taxa and thalattosuchians as non-neosuchians. Beginning with published unscaled phylogenetic topologies representing each of these hypotheses, we first manually added taxa present in our dataset but not in previous analyses [4]. We then time-calibrated each of these topologies using the fossilized birth–death model in MrBayes [49]. Fossil taxa were assigned

uniform temporal ranges based on known first and last occurrence dates using occurrence data derived from a recent phylogenetic analysis [4]. Models were run for 100 000 000 generations and sampling every 1000 generations with the first 50% of runs as burn-in. All model parameters and clock priors used default settings recommended from the literature [50]. Finally, we summarized the posterior distribution of trees by calculating maximum clade credibility trees for each topology.

Although using fossilized birth–death tip dating in this way is common practice for time-calibrating trees containing extinct taxa and has previously been used to date Crocodylomorpha phylogenies [4], we recovered divergence dates among extant taxa that are much more recent than estimates based on molecular evidence [51]. To be able to evaluate the effects of different dating approaches, we created a fourth phylogenetic hypothesis by modifying topology (C) to have divergence dates among extant taxa matching the mean node dates from a recent mitogenome-based phylogeny [51]. All four dated topologies (electronic supplementary material, figure S4) were used for all subsequent analyses and are referred to throughout the manuscript as follows: tree 1: molecular evidence scaled version of topology (C), tree 2: topology (A), tree 3: topology (B) and tree 4: topology (C).

## (d) Phenotypic variation

We visualized phenotypic variation using PCA. The first 16 PC axes represent 95% of the cumulative variation (figure 1). Phylogenetic signal was quantified using the multivariate extension of Blomberg's K using the function 'physignal' in the R package 'geomorph' v. 3.3.1 [34]. The effects of diet and body size were evaluated using Procrustes linear regression and Procrustes phylogenetic generalized least-squares regression [52]. For each regression, significance was evaluated using 999 iterations. We compared disparity between extant and extinct taxa and between brevirostrine crocodylians and all other taxa using the 'morphol.disparity' function in 'geomorph' [53]. We tested whether there is significant convergence in longirostrine taxa (*Cricosaurus*, *Gavialis*, *Mecistops*, *Pelagosaurus*, *Pholidosaurus*, *Tomistoma*) using the 'search.conv' function in the R package 'RRphylo' v. 2.4.4, calculating the angle between phenotypic vectors of species to identify convergent evolution [30]. To examine the relationship between evolutionary integration and morphological disparity across cranial regions, we calculated within-module integration using a likelihood-based approach, EMMLi [54]. For each phylogeny, we calculated phylogenetic independent contrasts of shape and used these values as input data for EMMLi. We extracted within-module integration values and plotted these against within-module Procrustes variance (electronic supplementary material, figure S5).

## (e) Rates analyses

We examined the tempo of skull evolution through time by fitting evolutionary models to the data for each phylogenetic hypothesis using BAYESTRAITS v. 3.2 (http://www.evolution.rdg.ac.uk/, [55]). BAYESTRAITS uses Bayesian inference to fit evolutionary models given a phylogenetic topology and phenotypic trait data, with model fit compared with Bayes factor we tested three models: (i)

all lineages evolve under a Brownian motion model and share a single rate of evolution (single rate Brownian motion), (ii) all lineages evolve at the same rate and are under selection, being drawn to an adaptive optimum ('single rate OU') and (iii) all lineages evolve under a Brownian motion model, but the rate of evolution varies across lineages and taxa (variable-rates Brownian motion). Under the variable-rates Brownian motion model, the rate of evolution on each branch is estimated by the model so that it is possible to compare relative rates among clades (e.g. do alligatoroids evolve slower or faster than crocodyloids?).

We first reduced the dimensionality of the data by subjecting the Procrustes-aligned coordinate configurations to phylogenetic PCA [56]. We used the PC scores for those axes representing a cumulative 95% of the total variance as input for BAYESTRAITS. We repeated each analysis for each phylogeny, using 100 000 000 iterations and a burn-in of 12 500 000 iterations. Each analysis was carried out twice and convergence of Markov chains confirmed using Gelman and Rubin's convergence diagnostic test statistic, implemented in 'coda' v. 0.19–3 [57]. For each phylogeny, we compared the likelihood of each model using Bayes factors (electronic supplementary material, figure S3) using the R package 'BTprocessR' (github.com/hferg/BTprocessR). Variable-rate Brownian motion is strongly favoured over single rate models (Bayes factor greater than 10) for each phylogeny.

We calculated the evolutionary rate for each cranial region using the multivariate rate metric, $\sigma_{mult}$ [58]. Although this approach assumes homogeneous rates across the tree, it has the advantage of explicit hypothesis testing (i.e. whether two modules evolve at significantly different rates). For each time-scaled topology, we using the 'compare.multi.evol.rates' function in the R package 'geomorph' to perform pairwise comparisons of rate between modules [53]. To examine how rates of evolution vary across the skull in detail, we calculated $\sigma_{mult}$ for each landmark using the R package 'hot.dots' (www.github.com/rnfelice/hot.dots).

Data accessibility. Three-dimensional mesh files of all specimens are archived at phenome10k.org (contingent on museum copyright policies). Code and data are available at www.github.com/rnfelice/Croc_Skulls/.

Authors' contributions. R.N.F.: Conceptualization, data curation, formal analysis, investigation, methodology, visualization, writing—original draft, writing—review and editing; D.P.: resources, writing—review and editing; A.G.: conceptualization, data curation, funding acquisition, investigation, methodology, project administration, resources, visualization, writing—original draft, writing—review and editing. All authors gave final approval for publication and agreed to be held accountable for the work performed therein.

Competing interests. We declare we have no competing interests.

Funding. This research was funded by European Research Council grant no. STG-2014-637171 (to A.G.).

Acknowledgements. We thank the museum curators and staff that facilitated access to these specimens: Michael Brett-Surman, Patrick Campbell, Martin Ezcurra, Anthony Herrel, Patricia Holroyd, David Kizirian (AMNH), Christine Lefevre, Alan Resetar, José Rosado and Alejo Scarano. Patrick O'Connor, Keegan Melstrom, Roland Sookias and Dan Paluh contributed anatomical reference material and scan data. Thanks to Akinobu Watanabe for feedback on an early version of this manuscript.

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
