## [Peer Review File · Proceedings of the Royal Society B: Biological Sciences]

Review History

RSPB-2020-2606.R0 (Original submission)

Review form: Reviewer 1

Recommendation

Accept with minor revision (please list in comments)

Scientific importance: Is the manuscript an original and important contribution to its field?

Excellent

General interest: Is the paper of sufficient general interest?

Good

Quality of the paper: Is the overall quality of the paper suitable?

Excellent

Is the length of the paper justified?

Yes

Should the paper be seen by a specialist statistical reviewer?

No

Do you have any concerns about statistical analyses in this paper? If so, please specify them explicitly in your report.

No

It is a condition of publication that authors make their supporting data, code and materials available - either as supplementary material or hosted in an external repository. Please rate, if applicable, the supporting data on the following criteria.

Is it accessible?

Yes

Is it clear?

Yes

Is it adequate?

Yes

Do you have any ethical concerns with this paper?

No

Comments to the Author

See attached file. (See Appendix A)

Review form: Reviewer 2 (Steven Salisbury)

Recommendation

Accept with minor revision (please list in comments)

Scientific importance: Is the manuscript an original and important contribution to its field?

Good

General interest: Is the paper of sufficient general interest?

Marginal

Quality of the paper: Is the overall quality of the paper suitable?

Good

Is the length of the paper justified?

Yes

Should the paper be seen by a specialist statistical reviewer?

Yes

Do you have any concerns about statistical analyses in this paper? If so, please specify them explicitly in your report.

No

It is a condition of publication that authors make their supporting data, code and materials available - either as supplementary material or hosted in an external repository. Please rate, if applicable, the supporting data on the following criteria.

Is it accessible?

Yes

Is it clear?

Yes

Is it adequate?

Yes

Do you have any ethical concerns with this paper?

No

Comments to the Author

I should state straight up that I do not understand exactly how high-dimensional 3D geometric morphometrics works. But I would say that I am very familiar with the crocodyliform skull and crocodyliform evolution (phylogenetics, biomechanics, feeding behaviour, locomotion, palaeobiogeography, etc) and ecology.

Despite my expertise, I found this manuscript very hard to penetrate. The language used, and the style in which key findings and insights are phrased, is very unfamiliar to me. There are too many assumptions made about the level of understanding that the reader has of this particular field (high definition 3D geometric morphometrics and Bayesian evolutionary modelling, etc). So the biggest issue for me is would this paper appeal to a broader scientific audience? I'm not saying that I am by any means representative of all fossil croc experts, but if I struggled with it, how will someone who is not a fossil croc expert take it? Will it appeal to them?

Is this paper too 'niche'?

At the moment, it feels like it is. I'm just being honest. I'm sure there are lots of evolutionary biologists out there who would be really interested in this article. I'm sure people who specialise in high definition 3D geometric morphometrics will love it. But that's not everyone.

I feel that if you made an effort to make the language used in this manuscript a little more easy to digest, it would appeal to a much broader audience. As it is now, I don't feel like it has a lot of broad appeal. It's just too technical and field-specific.

I have made lots of annotations on the two attached documents (pdf of the main text and a word doc of the SI). In addition to the over-arching comments above, some other things that came up include:

1) Australasian and Ocanian endemics

One of the main findings of your study is that Southeast Asian and Oceanian species of *Crocodylus*, have particularly high rates of evolution, despite exhibiting limited variation relative to extinct forms. In exploring this finding, you make reference to group of 'Australasian endemics' (p. 17 first paragraph). Are you referring to the Australasian species of *Crocodylus* (*C. jonhstoni* and *C. novaeguineae* are the only ones; *C. porosus* occurs throughout the Indo-Pacific) that are in your phylogenies or to extinct endemic crocodyoids (mekosuchines)? I assume it's the former, but this should be made clear.

What about *Crocodylus halli*?

Are you using 'Australasia' in the same sense that you use 'Oceania' in the abstract?

I note that you have included *Crocodylus raninus* in your analysis. As far as I'm aware, *C. raninus* is described from a skull and two preserved juveniles with no known extant population. On what basis are you considering it a distinct species? Given how similar it likely is to *C. porosus* and *C. novaeguineae* (and may even be synonymous), is it not surprising that appears to have diverged so recently in your phylogenies from these other *Crocodylus* spp.?

2) Ontogenetic allometry and intraspecific variation

One part of the analysis that concerned me was the fact that only a single specimen was used for each taxon. It is well known how much ontogenetic allometry occurs in crocodylians, independent of size. As crocs grow, various parts of the skull show allometric growth, while others are isometric. The skull of 1.5m juvenile *Crocodylus porosus* looks very different to that of a 5.5 m adult male, which also looks very different to the skull of a 2.5 m adult female. But they all belong to individuals within the same species. Crocs also have continual growth, but growth can be influenced by a range of factors. Two crocs of the same age (and level of maturity) can be very different sizes. The end result is a HUGE amount of morphological variability in skull shape within each species.

I'm sure you are aware of these issues, but it doesn't look as if this has been taken into account at all because only one specimen of unknown(?) ontogenetic stage is listed for each species used in the analyses. Are the specimens from wild caught or captive raised individuals? This will also affect how the data are interpreted. I assume that you have used what was a large (mature) individual for each specimen, but this is not stated. Where would juvenile or gerontic individuals sit within the various analyses?

This feels like a major shortcoming of the analysis, but is not even mentioned. How do you account for intraspecific ontogenetic variation in your dataset?

Related to this comment, within species, diet and ecology change with size (and usually maturity). Juvenile crocs eat small things (mainly invertebrates, small vertebrates, etc), and their diet changes as they grow. Some species such as *C. porosus* have different diets as adults depending on habitat segregation, some being more dependent on terrestrial prey while others consume mainly aquatic prey. It feels as if you are simplifying the dietary preferences of extant taxa to fit your preferred mode of analysis, but at the same time going to great lengths to add 'complexity' to how you analyse and interpret skull shape within individuals and differences in skull shape between species.

3) Minor quibbles: Your analysis includes *Pristichampsus vorax*, but *Pristichampsus* was placed in *Boverisuchus* in 2013. In Fig. 1 you include *Pholidosaurus perbeckensis*, but there is no mention of a specimen for this species in the specimen list in the Supp Info, only reference to *Pholidosaurus* sp.

For other comments, please refer to my annotations on the attached documents.

I want to like this manuscript, but I had a hard time understanding exactly how the analyses were conducted and what the results meant. I think this is mainly because of the language that is used, and the assumption that the reader is well versed in high resolution 3D geometric morphometrics and Bayesian evolutionary modelling. If you rephrase some of the text to make it more comprehensible to a broader audience, your manuscript will be much better received and the results more broadly appreciated. Once you do that, I would be happy to endorse it for publication in Proc B of the Royal Society.

Sincerely,
Steve Salisbury

Decision letter (RSPB-2020-2606.R0)

08-Dec-2020

Dear Dr Felice:

I am writing to inform you that your manuscript RSPB-2020-2606 entitled "Complex macroevolutionary dynamics underly the evolution of the crocodyliform skull" has, in its current form, been rejected for publication in Proceedings B.

This action has been taken on the advice of referees, who have recommended that substantial revisions are necessary. With this in mind we would be happy to consider a resubmission, provided the comments of the referees are fully addressed. However please note that this is not a provisional acceptance.

Sincerely,
Dr Daniel Costa
mailto: proceedingsb@royalsociety.org

Associate Editor
Board Member: 1
Comments to Author:

Both reviewers provided positive reviews but brought up some substantial issues. Rev 1 sought greater clarity on writing to ensure that readers were aware of the meaning of essential terms. S/he also thought that that your molecular evidence contradicted 3 of your 4 phylogenetic trees with respect to the crown-clade branch lengths and that this might bias the rate analysis. Rev 2 had suggestions about how to make the paper accessible to a greater number of potential readers - please go through the entire ms (and Rev 2's annotated file) to adjust the language to appeal to those not familiar with HD 3D morphometrics and Bayesian evolutionary modelling. S/he was also concerned about the large range in variability of skull shape within species and the relationship between dietary preferences and skull shape between species. This variability cannot be captured by the approach used in the paper - only 1 skull specimen for each taxon. The authors need to explain how intraspecific ontogenetic variation could affect their analyses. Rev 2 also has a number of additional comments and has provided annotations on the ms itself - as a separate file.

Reviewer(s)' Comments to Author:

Referee: 1

Comments to the Author(s)

See attached file.

Referee: 2

Comments to the Author(s)

I should state straight up that I do not understand exactly how high-dimensional 3D geometric morphometrics works. But I would say that I am very familiar with the crocodyliform skull and crocodyliform evolution (phylogenetics, biomechanics, feeding behaviour, locomotion, palaeobiogeography, etc) and ecology.

Despite my expertise, I found this manuscript very hard to penetrate. The language used, and the style in which key findings and insights are phrased, is very unfamiliar to me. There are too many assumptions made about the level of understanding that the reader has of this particular field (high definition 3D geometric morphometrics and Bayesian evolutionary modelling, etc).

So the biggest issue for me is would this paper appeal to a broader scientific audience? I'm not saying that I am by any means representative of all fossil croc experts, but if I struggled with it, how will someone who is not a fossil croc expert take it? Will it appeal to them?

Is this paper too 'niche'?

At the moment, it feels like it is. I'm just being honest. I'm sure there are lots of evolutionary biologists out there who would be really interested in this article. I'm sure people who specialise in high definition 3D geometric morphometrics will love it. But that's not everyone.

I feel that if you made an effort to make the language used in this manuscript a little more easy to digest, it would appeal to a much broader audience. As it is now, I don't feel like it has a lot of broad appeal. It's just too technical and field-specific.

I have made lots of annotations on the two attached documents (pdf of the main text and a word doc of the SI). In addition to the over-arching comments above, some other things that came up include:

1) Australasian and Ocanian endemics

One of the main findings of your study is that Southeast Asian and Oceanian species of *Crocodylus*, have particularly high rates of evolution, despite exhibiting limited variation relative to extinct forms. In exploring this finding, you make reference to group of 'Australasian endemics' (p. 17 first paragraph). Are you referring to the Australasian species of *Crocodylus* (*C. jonhstoni* and *C. novaeguineae* are the only ones; *C. porosus* occurs throughout the Indo-Pacific) that are in you phylogenies or to extinct endemic crocodyoids (mekosuchines)? I assume it's the former, but this should be made clear.

What about *Crocodylus halli*?

Are you using 'Australasia' in the same sense that you use 'Oceania' in the abstract?

I note that you have included *Crocodylus raninus* in your analysis. As far as I'm aware, *C. raninus* is described from a skull and two preserved juveniles with no known extant population. On what basis are you considering it a distinct species? Given how similar it likely is to *C. porosus* and *C. novaeguineae* (and may even be synonymous), is it not surprising that appears to have diverged so recently in your phylogenies from these other *Crocodylus* spp.?

2) Ontogenetic allometry and intraspecific variation

One part of the analysis that concerned me was the fact that only a single specimen was used for each taxon. It is well known how much ontogenetic allometry occurs in crocodylians, independent of size. As crocs grow, various parts of the skull show allometric growth, while others are isometric. The skull of 1.5m juvenile *Crocodylus porosus* looks very different to that of a 5.5 m adult male, which also looks very different to the skull of a 2.5 m adult female. But they all belong to individuals within the same species. Crocs also have continual growth, but growth can be influenced by a range of factors. Two crocs of the same age (and level of maturity) can be very different sizes. The end results is a HUGE amount of morphological variability in skull shape within each species.

I'm sure you are aware of these issues, but it doesn't look as if this has taken this into account at all because only one specimen of unknown(?) ontogenetic stage is listed for each species used in the analyses. Are the specimens from wild caught or captive raised individuals? This will also affect how the data are interpreted. I assume that you have used what was a large (mature) individual for each specimen, but this is not stated. Where would juvenile or gerontic individuals sit within the various analyses?

This feels like a major shortcoming of the analysis, but is not even mentioned. How do you account for intraspecific ontogenetic variation in your dataset?

Related to this comment, within species, diet and ecology change with size (and usually maturity). Juvenile crocs eat small things (mainly invertebrates, small vertebrates, etc), and their diet changes as they grow. Some species such as *C. porosus* have different diets as adults depending on habitat segregation, some being more dependent on terrestrial prey while others consume mainly aquatic prey. It feels as if you are simplifying the dietary preferences of extant taxa to fit your preferred mode of analysis, but at the same time going to great lengths to add 'complexity' to how you analyse and interpret skull shape within individuals and differences in skull shape between species.

3) Minor quibbles: Your analysis includes *Pristichampsus vorax*, but *Pristichampsus* was placed in *Boverisuchus* in 2013. In Fig. 1 you include *Pholidosaurus perbeckensis*, but there is no mention of a specimen for this species in the specimen list in the Supp Info, only reference to *Pholidosaurus* sp.

For other comments, please refer to my annotations on the attached documents.

I want to like this manuscript, but I had a hard time understanding exactly how the analyses were conducted and what the results meant. I think this is mainly because of the language that is used, and the assumption that the reader is well versed in high resolution 3D geometric morphometrics and Bayesian evolutionary modelling. If you rephrase some of the text to make it more comprehensible to a broader audience, your manuscript will be much better received and the results more broadly appreciated. Once you do that, I would be happy to endorse it for publication in Proc B of the Royal Society.

Sincerely,
Steve Salisbury

Author's Response to Decision Letter for (RSPB-2020-2606.R0)

See Appendix B.

RSPB-2021-0919.R0

Review form: Reviewer 1

Recommendation

Accept as is

Scientific importance: Is the manuscript an original and important contribution to its field?

Good

General interest: Is the paper of sufficient general interest?

Good

Quality of the paper: Is the overall quality of the paper suitable?

Excellent

Is the length of the paper justified?

Yes

Should the paper be seen by a specialist statistical reviewer?

No

Do you have any concerns about statistical analyses in this paper? If so, please specify them explicitly in your report.

No

It is a condition of publication that authors make their supporting data, code and materials available - either as supplementary material or hosted in an external repository. Please rate, if applicable, the supporting data on the following criteria.

Is it accessible?

Yes

Is it clear?

Yes

Is it adequate?

Yes

Do you have any ethical concerns with this paper?

No

Comments to the Author

As I said in my prior review, I find this to be a very interesting and important paper. Having gone through the revised manuscript I feel that the authors have satisfactorily addressed all of my prior concerns and suggestions. I recommend acceptance once some outstanding nomenclatural issues are dealt with.

Given the use of the molecular topology for Crocodylia, there are some nomenclatural issues in the paper that must be addressed.

Under the molecular topology, Brevirostres is a junior synonym of Crocodylia. Authors should use Crocodylia when discussing clades based on this tree.

see lines 124, 243, 264, Figure 1 caption

Line 126: "stem-crocodylian". Not sure what they mean here. The only non-crocodyloid non-alligatoroid crocodylian in the the tree is Boverisuchus. What do they mean by "stem-crocodylian"

Line 273 and 277: Authors use "Crocodyloidea" here. Under the molecular topology Crocodyloidea includes Gavialis, so I'm not sure they actually mean Crocodyloidea. They probably mean Crocodylidae, as this would exclude Gavialis, *C. affinis*, and Prodiplacynodon.

Regarding Figure 1.

First, the "A" label is missing from part A of the figure. Second, the orange-ish color used for Crocodylidae in Figure 1A looks identical to the "terrestrial" color in parts B and C. This is pretty confusing as no crocodylid is terrestrial. Its not immediately apparent that the color scheme shifts between the two parts of the figure.

"Pristichampsus" should be "Boverisuchus".

Pristichampsus vorax, Prodiplacynodon sp., and Crocodylus affinis are colored as "Crocodylidae" but are not members of this clade. Prodiplacynodon and Crocodylus affinis are crocodyloids but not crocodylids. Pristichampsus is not even a Crocodyloid.

There are a number of "Extant Alligatoroidea" dots in Figure 1A that do not have species names associated with them. There is one "Extant Crocodylidae" dot next to Leidysuchus that is also not labeled. Were these all intentional?

Why give Kaprosuchus the Mahajangasuchidae label as opposed to Notosuchia? It is deeply nested in Notosuchia and Kaprosuchus/Mahajangasuchidae is never mentioned in the text?

Figure 1 caption: "PC axis 2 separates the extant Brevirostres (alligators, caimans, and crocodiles) from stem groups (C)." This is another confusing use of Brevirostres and an imprecise use of "stem groups". Are the authors including Gavialis as a "stem group". And again, Brevirostres is synonymous with Crocodylia under the molecular topology.

Figure 3.

The "Crocodyloidea" black bar does not overlap with the correct taxa in the cladogram.

The "Alligatoroidea" black bar does not overlap with the correct taxa in the cladogram.

The "Notosuchia" black bar does not overlap with the correct taxa in the cladogram.

Given all of these nomenclatural issues, I suggest the authors include a brief "nomenclature" section early in the paper (or less ideally, in the supplement) where they provide the names and definitions they are using. I would further recommend that if the author want to refer to the "crocodyloids and alligatoroids that aren't gavialids" they could make clear in this new Nomenclature section that they will use the term "brevirostrine crocodylians" to refer to this paraphyletic group based on molecular topologies as it distinguishes a group of crocodylians with similar snout shapes that are recovered as the clade Brevirostres in the morphological trees.

Review form: Reviewer 2

Recommendation

Accept with minor revision (please list in comments)

Scientific importance: Is the manuscript an original and important contribution to its field?

Excellent

General interest: Is the paper of sufficient general interest?

Good

Quality of the paper: Is the overall quality of the paper suitable?

Excellent

Is the length of the paper justified?

Yes

Should the paper be seen by a specialist statistical reviewer?

Yes

Do you have any concerns about statistical analyses in this paper? If so, please specify them explicitly in your report.

No

It is a condition of publication that authors make their supporting data, code and materials available - either as supplementary material or hosted in an external repository. Please rate, if applicable, the supporting data on the following criteria.

Is it accessible?

Yes

Is it clear?

Yes

Is it adequate?

Yes

Do you have any ethical concerns with this paper?

No

Comments to the Author

I think you've done a great job addressing the issues that were raised with the initial submission. I very much appreciated the way the language of the paper has been tempered to make it more accessible to non-specialists. THANK YOU! I have the response to the reviews and re-read the manuscript in its entirety, and found the entire experience much more enjoyable than I did the first time around. I made a few minor edits (mainly grammatical/typo related) to the attached word document, and added a few comments.

One thing I noticed this time relates to the use of the term 'snout'. In most instances, you seem to be using it in reference to the upper jaws or more specifically the maxillary rostrum, but you don't actually specify this. Later in the text you start talking about the mandible as being separate to the snout, and then you drop in a few 'rostrums'. Given how critical the 'snout' is to this manuscript, I think it's probably worth being a bit more precise with your terminology. Rostrum (= 'the snout') includes both the maxillary rostrum and the mandibular rostrum. The maxillary rostrum is formed by the ankylosis of the right and left premaxillary, maxillary, lacrimals and prefrontal bones, and to varying degrees the right and left palatal bones and the rostral process of the frontal. The mandibular rostrum corresponds to the portion of the mandible formed by the union of the symphyseal segments of the right and left mandibular rami. Although this region of ankylosis of the mandibular rami is often called the 'mandibular symphysis', in a strict sense the symphysis refers only to the actual joint connecting the two bones.

Once these minor edits are accepted, I'd be happy to support publication of this article in Proc RSB. It's great to see it reach this point. Kind regards, Steve Salisbury

Decision letter (RSPB-2021-0919.R0)

15-Jun-2021

Dear Dr Felice

I am pleased to inform you that your manuscript RSPB-2021-0919 entitled "Complex macroevolutionary dynamics underly the evolution of the crocodyliform skull" has been accepted for publication in Proceedings B.

The referee(s) have recommended publication, but also suggest some minor revisions to your manuscript. Therefore, I invite you to respond to the referee(s)' comments and revise your manuscript. Because the schedule for publication is very tight, it is a condition of publication that you submit the revised version of your manuscript within 7 days. If you do not think you will be able to meet this date please let us know.

Sincerely,

Dr Daniel Costa

Reviewer(s)' Comments to Author:

Referee: 1

Comments to the Author(s).

As I said in my prior review, I find this to be a very interesting and important paper. Having gone through the revised manuscript I feel that the authors have satisfactorily addressed all of my prior concerns and suggestions. I recommend acceptance once some outstanding nomenclatural issues are dealt with.

Given the use of the molecular topology for Crocodylia, there are some nomenclatural issues in the paper that must be addressed.

Under the molecular topology, Breviostres is a junior synonym of Crocodylia. Authors should use Crocodylia when discussing clades based on this tree.

see lines 124, 243, 264, Figure 1 caption

Line 126: "stem-crocodylian". Not sure what they mean here. The only non-crocodyloid non-alligatoroid crocodylian in the tree is Boverisuchus. What do they mean by "stem-crocodylian"

Line 273 and 277: Authors use "Crocodyloidea" here. Under the molecular topology Crocodyloidea includes Gavialis, so I'm not sure they actually mean Crocodyloidea. They probably mean Crocodylidae, as this would exclude Gavialis, *C. affinis*, and Prodiplacynodon.

Regarding Figure 1.

First, the "A" label is missing from part A of the figure. Second, the orange-ish color used for Crocodylidae in Figure 1A looks identical to the "terrestrial" color in parts B and C. This is pretty confusing as no crocodylid is terrestrial. It's not immediately apparent that the color scheme shifts between the two parts of the figure.

"Pristichampsus" should be "Boverisuchus".

Pristichampsus vorax, *Prodiplacynodon* sp., and *Crocodylus affinis* are colored as "Crocodylidae" but are not members of this clade. *Prodiplacynodon* and *Crocodylus affinis* are crocodyloids but not crocodylids. *Pristichampsus* is not even a Crocodyloid.

There are a number of "Extant Alligatoroidea" dots in Figure 1A that do not have species names associated with them. There is one "Extant Crocodylidae" dot next to *Leidyosuchus* that is also not labeled. Were these all intentional?

Why give *Kaprosuchus* the Mahajangasuchidae label as opposed to Notosuchia? It is deeply nested in Notosuchia and *Kaprosuchus*/*Mahajangasuchidae* is never mentioned in the text?

Figure 1 caption: "PC axis 2 separates the extant Brevirostres (alligators, caimans, and crocodiles) from stem groups (C)." This is another confusing use of Brevirostres and an imprecise use of "stem groups". Are the authors including *Gavialis* as a "stem group". And again, Brevirostres is synonymous with Crocodylia under the molecular topology.

Figure 3.

The "Crocodyloidea" black bar does not overlap with the correct taxa in the cladogram.

The "Alligatoroidea" black bar does not overlap with the correct taxa in the cladogram.

The "Notosuchia" black bar does not overlap with the correct taxa in the cladogram.

Given all of these nomenclatural issues, I suggest the authors include a brief "nomenclature" section early in the paper (or less ideally, in the supplement) where they provide the names and definitions they are using. I would further recommend that if the author wants to refer to the "crocodyloids and alligatoroids that aren't gavialids" they could make clear in this new Nomenclature section that they will use the term "brevirostrine crocodylians" to refer to this paraphyletic group based on molecular topologies as it distinguishes a group of crocodylians with similar snout shapes that are recovered as the clade Brevirostres in the morphological trees.

Referee: 2

Comments to the Author(s).

I think you've done a great job addressing the issues that were raised with the initial submission.

I very much appreciated the way the language of the paper has been tempered to make it more accessible to non-specialists. THANK YOU! I have the response to the reviews and re-read the manuscript in its entirety, and found the entire experience much more enjoyable than I did the first time around. I made a few minor edits (mainly grammatical/typo related) to the attached word document, and added a few comments.

One thing I noticed this time relates to the use of the term 'snout'. In most instances, you seem to be using it in reference to the upper jaws or more specifically the maxillary rostrum, but you don't actually specify this. Later in the text you start talking about the mandible as being separate to the snout, and then you drop in a few 'rostrums'. Given how critical the 'snout' is to this manuscript, I think it's probably worth being a bit more precise with your terminology. Rostrum (= 'the snout') includes both the maxillary rostrum and the mandibular rostrum. The maxillary rostrum is formed by the ankylosis of the right and left premaxillary, maxillary, lacrimals and prefrontal bones, and to varying degrees the right and left palatal bones and the rostral process of the frontal. The mandibular rostrum corresponds to the portion of the mandible formed by the union of the symphyseal segments of the right and left mandibular rami. Although this region of

ankylosis of the mandibular rami is often called the 'mandibular symphysis', in a strict sense the symphysis refers only to the actual joint connecting the two bones.

Once these minor edits are accepted, I'd be happy to support publication of this article in Proc RSB. It's great to see it reach this point. Kind regards, Steve Salisbury

Author's Response to Decision Letter for (RSPB-2021-0919.R0)

See Appendix C.

Decision letter (RSPB-2021-0919.R1)

16-Jun-2021

Dear Dr Felice

I am pleased to inform you that your manuscript entitled "Complex macroevolutionary dynamics underly the evolution of the crocodyliform skull" has been accepted for publication in Proceedings B.

Data Accessibility section

Open Access

Paper charges

Sincerely,
Proceedings B
<mailto:proceedingsb@royalsociety.org>

Appendix A

The authors present a novel 3D GMM analysis of cranial shape in a large set of crocodyliforms. Using their high dimensional data-set they are able to quantify phenotypic disparity and degree of integration within cranial elements for the clade. Evolutionary modelling reveals some unexpected findings such as high rates of shape evolution with modern crocodyloids. In total I think this is a very interesting paper, the methods are appropriate and well applied, and the results are significant and of broad interest. I would recommend acceptance following some minor revisions.

SPECIFIC COMMENTS and/or SUGGESTIONS:

Line 19: Saying that crocodyliforms experienced bursts of rapid evolutionary change at the origin of major clades, overstates the results a bit. The majority of the estimated rate shifts within the phylogenies examined occurred within the crown-group. Authors should consider revising the wording to be a bit more clear as to *which* major clades.

Line 88: With respect to thoughts on why secondary palates formed, I would suggest the authors also look to Langston (1973), Busbey (1995), Daniel and McHenry (2001), and Turner and Buckley (2008) for additional references with regards to this anatomy. Even in eusuchians some folks think it is more a structural response to a dorsoventrally compressed snout.

Line 168: This is the first mention of “module” in the paper. Perhaps you want to be explicit as what you consider a module. I’m not sure it’s safe to assume 1) that each reader will be familiar with the ideas behind modularity and integration and/or 2) what part of the phenotype you are considering to be discrete modules. In this case, it seems like you are using “module” pretty much interchangeable with “individual skull element” (perhaps with the exception of occiput). Even the more general “jaw joint” seems to be referring to the quadrate (perhaps just the condylar portion??). I think explicitness here is important.

Line 175: Sort of the same point here. This is the first mention of “integration” without giving the reader some background on what the biological meaning of this is.

Lines 217-221: Because of its role in supporting rotational forces, an extensive bony secondary palate is probably an evolutionary prerequisite to lots of dorsoventral snout compression. (see comment and citations above). Absence of this morphology in more crocodyliform clades outside of Eusuchia might explain the rarity of this phenotype in the group...just a thought...

Line 246: “rapidly exploring a limited range of phenotypes”. Perhaps this is too nitpicky but “exploring” and “limited range” don’t seem to match well. Is there another descriptor the authors could consider for what they see crocodyloid doing in their data-set. “Rapidly fluctuating within a limited range” perhaps? Exploring sort of evokes a sense of finding something new when in this case it seems like they just repeatedly bounce around a small set of shapes.

Time-scaling the phylogenetic trees and the authors preferred tree for figure 2: The method the authors use to time scale their tree is appropriate in my opinion. However, I wonder if the authors could clarify why they have chosen the specific tree they did for depicting their results in Figure 2. As they note, 3 of the 4 trees used in their analyses do not place crown group divergences at times consistent with molecular evidence. This raises a concern for me as it would appear that molecular evidence therefore contradicts 3 of their 4 trees with respect to the crown-clade branch lengths. My concern is that this may bias the rate analysis and lead to

appearance of comparatively fast rates in tips with very very short internodes (like in crocodyloids). Tree 4 in their study certainly seems to spread the rate across the tree a bit more (while not erasing the elevated rates in crocodyloids) and suggesting nearly comparably high rates in notosuchians.

None of this I think alters the significance or impact of the paper but perhaps the authors could address their phylogenetic choices in the methods section more fully or consider using Tree 4 as their primary tree as it is the one most consistent with all available data vis a vis branch lengths.

CITATIONS

Busbey, A. B. 1995. The structural consequences of skull flattening in crocodylians; pp. 173–192 in J. J. Thomason (ed), *Functional morphology in Vertebrate Paleontology*, Cambridge University Press, Cambridge.

Langston, W. 1973. The crocodylian skull in historical perspective; pp. 263–284 in C. Gans and T. S. Parsons (eds.), *Biology of the Reptilia*, vol. 4, Academic Press, London.

Daniel, W. J. T. and C. McHenry. 2001. Bite force to skull stress correlation—modeling the skull of *Alligator mississippiensis*; pp. 135–143 in G. C. Grigg, F. Seebacher and C. E. Franklin (eds.), *Crocodylian biology and evolution*. Surrey Beatty and Sons, Chipping Norton.

Turner, A.H. and G. A. Buckley. 2008. *Mahajangasuchus insignis* (Crocodyliformes: Mesoeucrocodylia) cranial anatomy and new data on the origin of the eusuchian-style palate. *Journal of Vertebrate Paleontology* 28(2): 382–408.

Appendix B

Response to Reviewers:

Associate Editor

Board Member: 1

Comments to Author:

Both reviewers provided positive reviews but brought up some substantial issues. Rev 1 sought greater clarity on writing to ensure that readers were aware of the meaning of essential terms. S/he also thought that that your molecular evidence contradicted 3 of your 4 phylogenetic trees with respect to the crown-clade branch lengths and that this might bias the rate analysis. Rev 2 had suggestions about how to make the paper accessible to a greater number of potential readers – please go through the entire ms (and Rev 2's annotated file) to adjust the language to appeal to those not familiar with HD 3D morphometrics and Bayesian evolutionary modelling. S/he was also concerned about the large range in variability of skull shape within species and the relationship between dietary preferences and skull shape between species. This variability cannot be captured by the approach used in the paper - only 1 skull specimen for each taxon. The authors need to explain how intraspecific ontogenetic variation could affect their analyses. Rev 2 also has a number of additional comments and has provided annotations on the ms itself – as a separate file.

Reviewer(s)' Comments to Author:

Referee: 1

The authors present a novel 3D GMM analysis of cranial shape in a large set of crocodyliforms.

Using their high dimensional data-set they are able to quantify phenotypic disparity and degree of integration within cranial elements for the clade. Evolutionary modelling reveals some unexpected findings such as high rates of shape evolution with modern crocodyloids. In total I think this is a very interesting paper, the methods are appropriate and well applied, and the results are significant and of broad interest. I would recommend acceptance following some minor revisions.

Thank you for your very positive comments and helpful suggestions on our manuscript.

SPECIFIC COMMENTS and/or SUGGESTIONS:

Line 19: Saying that crocodyliforms experienced bursts of rapid evolutionary change at the origin of major clades, overstates the results a bit. The majority of the estimated rate shifts within the phylogenies examined occurred within the crown-group. Authors should consider revising the wording to be a bit more clear as to which major clades.

-RESPONSE: We have changed this line to more accurately summarise the results. It now reads: “Evolutionary modelling demonstrates that the pace of crocodyliform cranial evolution is initially high, particularly in the extinct Notosuchia, but slows near the base of Neosuchia, with a late burst of rapid evolution in crown-group crocodiles”

Line 88: With respect to thoughts on why secondary palates formed, I would suggest the authors also look to Langston (1973), Busbey (1995), Daniel and McHenry (2001), and Turner and Buckley (2008) for additional references with regards to this anatomy. Even in eusuchians some folks think it is more a structural response to a dorsoventrally compressed snout.

-Response: Great point. Lines 76-80 now read: “For example, different groups have independently acquired a derived secondary palate formed by the palatine and pterygoid bones, which in neosuchians is thought to be an adaptation for semi-aquatic life [20] and for resisting the torsional and compressive forces experienced during feeding [3,21].”

Line 168: This is the first mention of “module” in the paper. Perhaps you want to be explicit as what you consider a module. I’m not sure it’s safe to assume 1) that each reader will be familiar with the ideas behind modularity and integration and/or 2) what part of the phenotype you are considering to be discrete modules. In this case, it seems like you are using “module” pretty much interchangeable with “individual skull element” (perhaps with the exception of occiput). Even the more general “jaw joint” seems to be referring to the quadrate (perhaps just the condylar portion??). I think explicitness here is important.

-Response: we have added an additional paragraph to the introduction (Lines 102-110) to introduce the concepts of modularity and integration more clearly. We explain on line 199 that we are using the term “jaw joint” to refer to the articular surface of the quadrate.

. Evolutionary modelling demonstrates that as a whole, crocodyliforms have experienced heterogenous rates of evolution through time, with notable bursts of rapid evolution in crown-group crocodiles (Crocodylioidea) and in the ecologically diverse Cretaceous clade Notosuchia

Line 175: Sort of the same point here. This is the first mention of “integration” without giving the reader some background on what the biological meaning of this is.

-Response: See response to the previous comment. We have defined and contextualized this term more clearly at the end of the introduction

Lines 217-221: Because of its role in supporting rotational forces, an extensive bony secondary palate is probably an evolutionary prerequisite to lots of dorsoventral snout compression. (see comment and citations above). Absence of this morphology in more crocodyliform clades outside of Eusuchia might explain the rarity of this phenotype in the group...just a thought...

-Response: Interesting thought and definitely worthy of further inquiry. There may be a possibility of modelling trait-dependent character state transitions to test this hypothesis, but it would really beg the inclusion of other taxa with this palate type (eg shartegosuchids).

Line 246: “rapidly exploring a limited range of phenotypes”. Perhaps this is too nitpicky but “exploring” and “limited range” don’t seem to match well. Is there another descriptor the authors could consider for what they see crocodyloid doing in their data-set. “Rapidly fluctuating within a limited range” perhaps? Exploring sort of evokes a sense of finding something new when in this case it seems like they just repeatedly bounce around a small set of shapes.

Response: this now reads “Thus, extant lineages are not in evolutionary stasis but rather have rapidly fluctuated within a limited region of morphospace,” (line 22-23)

Time-scaling the phylogenetic trees and the authors preferred tree for figure 2: The method the authors use to time scale their tree is appropriate in my opinion. However, I wonder if the

authors could clarify why they have chosen the specific tree they did for depicting their results in Figure 2. As they note, 3 of the 4 trees used in their analyses do not place crown-group divergences at times consistent with molecular evidence. This raises a concern for me as it would appear that molecular evidence therefore contradicts 3 of their 4 trees with respect to the crown-clade branch lengths. My concern is that this may bias the rate analysis and lead to appearance of comparatively fast rates in tips with very very short internodes (like in crocodyloids). Tree 4 in their study certainly seems to spread the rate across the tree a bit more (while not erasing the elevated rates in crocodyloids) and suggesting nearly comparably high rates in notosuchians. None of this I think alters the significance or impact of the paper but perhaps the authors could address their phylogenetic choices in the methods section more fully or consider using Tree 4 as their primary tree as it is the one most consistent with all available data vis a vis branch lengths.

Response: We have reorganized the manuscript and figures to put the molecular-scaled tree as the primary tree figured in the manuscript text.

Referee: 2

Comments to the Author(s)

I should state straight up that I do not understand exactly how high-dimensional 3D geometric morphometrics works. But I would say that I am very familiar with the crocodyliform skull and crocodyliform evolution (phylogenetics, biomechanics, feeding behaviour, locomotion, palaeobiogeography, etc) and ecology.

Despite my expertise, I found this manuscript very hard to penetrate. The language used, and the style in which key findings and insights are phrased, is very unfamiliar to me. There are too many assumptions made about the level of understanding that the reader has of this particular field (high definition 3D geometric morphometrics and Bayesian evolutionary modelling, etc).

So the biggest issue for me is would this paper appeal to a broader scientific audience? I'm not saying that I am by any means representative of all fossil croc experts, but if I struggled with it, how will someone who is not a fossil croc expert take it? Will it appeal to them?

Is this paper too 'niche'?

At the moment, it feels like it is. I'm just being honest. I'm sure there are lots of evolutionary biologists out there who would be really interested in this article. I'm sure people who specialise in high definition 3D geometric morphometrics will love it. But that's not everyone.

Thank you for your feedback on the presentation of our work and suggestions for how to make it more accessible. We have revised the language in the introduction and discussion to clarify and emphasize the significance of these results to readers not versed in modern evolutionary biology methods. We have also added in an additional figure to highlight the novel perspective on crocodyliform cranial evolution provided by the work, which features “exploded skull” views showing the morphology of each cranial element on the extremes of the major axes of variation, which we think will have broad appeal and demonstrate some of the interesting features discussed in the manuscript.

With regard to whether this paper is too 'niche', we respectfully disagree. These findings will be of interest for any researcher interested in cranial evolution, constraint/convergence, and macroevolutionary tempo and mode. This journal has a long history of publishing outstanding papers that, like this one, use data collected

from crocodylians and their relatives to test evolutionary hypotheses [1-7]. As noted in the cover letter, in the few months since we received the decision on our manuscript, a paper using similar methods but a standard 2D dataset for crocodyliforms was submitted, reviewed, and published in this journal, again suggesting that the topic of this paper is of interest for the readership of *Proceedings B*. [7]

1. Bona P, Ezcurra MD, Barrios F, Fernandez Blanco MV. 2018 A new Palaeocene crocodylian from southern Argentina sheds light on the early history of caimanines. *Proceedings of the Royal Society B: Biological Sciences* 285, 20180843. (doi:[10.1098/rspb.2018.0843](https://doi.org/10.1098/rspb.2018.0843))
2. McCurry MR, Evans AR, Fitzgerald EMG, Adams JW, Clausen PD, McHenry CR. 2017 The remarkable convergence of skull shape in crocodylians and toothed whales. *Proceedings of the Royal Society B: Biological Sciences* 284, 20162348. (doi:[10.1098/rspb.2016.2348](https://doi.org/10.1098/rspb.2016.2348))
3. Morris ZS, Vliet KA, Abzhanov A, Pierce SE. 2019 Heterochronic shifts and conserved embryonic shape underlie crocodylian craniofacial disparity and convergence. *Proceedings of the Royal Society B: Biological Sciences* 286, 20182389. (doi:[10.1098/rspb.2018.2389](https://doi.org/10.1098/rspb.2018.2389))
4. Salisbury SW, Molnar RE, Frey E, Willis PMA. 2006 The origin of modern crocodyliforms: new evidence from the Cretaceous of Australia. *Proc. R. Soc. B.* 273, 2439–2448. (doi:[10.1098/rspb.2006.3613](https://doi.org/10.1098/rspb.2006.3613))
5. Stubbs TL, Pierce SE, Rayfield EJ, Anderson PSL. 2013 Morphological and biomechanical disparity of crocodile-line archosaurs following the end-Triassic extinction. *Proceedings of the Royal Society B: Biological Sciences* 280, 20131940–20131940. (doi:[10.1098/rspb.2013.1940](https://doi.org/10.1098/rspb.2013.1940))
6. Turner AH. 2004 Crocodyliform biogeography during the Cretaceous: evidence of Gondwanan vicariance from biogeographical analysis. *Proceedings of the Royal Society of London. Series B: Biological Sciences* 271, 2003–2009. (doi:[10.1098/rspb.2004.2840](https://doi.org/10.1098/rspb.2004.2840))
7. Stubbs TL, Pierce SE, Elsler A, Anderson PSL, Rayfield EJ, Benton MJ. 2021 Ecological opportunity and the rise and fall of crocodylomorph evolutionary innovation. *Proc. R. Soc. B.* 288, [rspb.2021.0069](https://doi.org/10.1098/rspb.2021.0069), 20210069. (doi:[10.1098/rspb.2021.0069](https://doi.org/10.1098/rspb.2021.0069))

I feel that if you made an effort to make the language used in this manuscript a little more easy to digest, it would appeal to a much broader audience. As it is now, I don't feel like it has a lot of broad appeal. It's just too technical and field-specific.

Response:

We have extensively revised our manuscript to make it more accessible for a broader audience. In addition to the overall text, we note that this comment also relates to this reviewer's comments on the supplemental figures. To address this, we have added additional detail to the legends and captions for the supplemental figures to increase clarity for non-specialist readers.

I have made lots of annotations on the two attached documents (pdf of the main text and a word doc of the SI). In addition to the over-arching comments above, some other things that came up include:

1) Australasian and Ocianian endemics

One of the main findings of your study is that Southeast Asian and Oceanian species of *Crocodylus*, have particularly high rates of evolution, despite exhibiting limited variation

relative to extinct forms. In exploring this finding, you make reference to group of 'Australasian endemics' (p. 17 first paragraph). Are you referring to the Australasian species of *Crocodylus* (*C. johnstoni* and *C. novaeguineae* are the only ones; *C. porosus* occurs throughout the Indo-Pacific) that are in your phylogenies or to extinct endemic crocodyloids (mekosuchines)? I assume it's the former, but this should be made clear.

Response: Thanks for pointing out the ambiguity here. This line has been changed to read "The position of this shift varies slightly across the four trees, but always comprises a clade of taxa that are found in Australia, southeast Asia, and the Indo-Pacific."

What about *Crocodylus halli*?

Are you using 'Australasia' in the same sense that you use 'Oceania' in the abstract?

Response: Thank you for pointing out the ambiguity here. We have changed the abstract to match the text in the discussion: "Surprisingly, modern crocodiles, especially Australian, southeast Asian, Indo-Pacific species, have high rates of evolution, despite exhibiting low variation."

I note that you have included *Crocodylus raninus* in your analysis. As far as I'm aware, *C. raninus* is described from a skull and two preserved juveniles with no known extant population. On what basis are you considering it a distinct species? Given how similar it likely is to *C. porosus* and *C. novaeguineae* (and may even be synonymous), is it not surprising that it appears to have diverged so recently in your phylogenies from these other *Crocodylus* spp.?

Response: Thank you for pointing this out. Given the uncertainty in the status of this taxon, we have removed *C. raninus* from our analyses

2) Ontogenetic allometry and intraspecific variation

One part of the analysis that concerned me was the fact that only a single specimen was used for each taxon. It is well known how much ontogenetic allometry occurs in crocodylians, independent of size. As crocs grow, various parts of the skull show allometric growth, while others are isometric. The skull of 1.5m juvenile *Crocodylus porosus* looks very different to that of a 5.5 m adult male, which also looks very different to the skull of a 2.5 m adult female. But they all belong to individuals within the same species. Crocs also have continual growth, but growth can be influenced by a range of factors. Two crocs of the same age (and level of maturity) can be very different sizes. The end result is a HUGE amount of morphological variability in skull shape within each species.

I'm sure you are aware of these issues, but it doesn't look as if this has been taken into account at all because only one specimen of unknown(?) ontogenetic stage is listed for each species used in the analyses. Are the specimens from wild caught or captive raised individuals? This will also affect how the data are interpreted. I assume that you have used what was a large (mature) individual for each specimen, but this is not stated. Where would juvenile or gerontic individuals sit within the various analyses?

This feels like a major shortcoming of the analysis, but is not even mentioned. How do you account for intraspecific ontogenetic variation in your dataset?

Response: The reviewer raises an important point about ontogenetic variation and continual growth in these animals, although we would argue that ontogenetic allometry, by definition, cannot be independent of size. We have added a section to the "Sampling" part of the Methods section explaining that we used wild-caught adult

specimens only. Intraspecific variability is an important factor in all comparative studies, but its impact on results must be considered in relation to the scale of analyses. Here, we are focused on large-scale patterns spanning crocodyliforms over ~200 million years. At this scale, intraspecific variation among adults in a species is minute compared to interspecific variation. We certainly agree that ontogenetic variation could mislead results, but as detailed in our expanded Sampling section, and as has been applied in other broad-scale crocodylomorph morphometric analyses (refs 1-4 below), we are using one adult specimen per taxon. Also of note is that another similar analysis of crocodyliform skull shape evolution published in this journal does not specify the age class of specimens [1].

1. Stubbs TL, Pierce SE, Elsler A, Anderson PSL, Rayfield EJ, Benton MJ. 2021 Ecological opportunity and the rise and fall of crocodylomorph evolutionary innovation. *Proc. R. Soc. B.* 288, rspb.2021.0069, 20210069. (doi:10.1098/rspb.2021.0069)
2. Godoy PL, Ferreira GS, Montefeltro FC, Vila Nova BC, Butler RJ, Langer MC. 2018 Evidence for heterochrony in the cranial evolution of fossil crocodyliforms. *Palaeontology* 61, 543–558. (doi:10.1111/pala.12354)
3. Drumheller SK, Wilberg EW. 2019 A synthetic approach for assessing the interplay of form and function in the crocodyliform snout. *Zoological Journal of the Linnean Society* (doi:10.1093/zoolinnean/zlz081)
4. Pierce, S. E., K. D. Angielczyk, and E. J. Rayfield. 2009. Morphospace occupation in thalattosuchian crocodylomorphs: skull shape variation, species delineation and temporal patterns. *Palaeontology* 52:1057–1097.

Related to this comment, within species, diet and ecology change with size (and usually maturity). Juvenile crocs eat small things (mainly invertebrates, small vertebrates, etc), and their diet changes as they grow. Some species such as *C. porosus* have different diets as adults depending on habitat segregation, some being more dependent on terrestrial prey while others consume mainly aquatic prey. It feels as if you are simplifying the dietary preferences of extant taxa to fit your preferred mode of analysis, but at the same time going to great lengths to add 'complexity' to how you analyse and interpret skull shape within individuals and differences in skull shape between species.

Response: We agree that ontogenetic variation in diet is an important aspect of crocodylomorph ecology, cranial function, and evolution, and this is an important and interesting phenomenon in many taxa. However, in this study, we are using only adult individuals in part to control for this complex axis of variation.

We disagree that we are simplifying dietary preferences. A recent in-depth review of dietary preferences in crocodylomorphs demonstrated that modern eusuchians show highly variable diets that include tetrapods and fish, putting them in to the “generalist” dietary category as adults (1). While there is always some degree of dietary variation across individuals in a species, and this is a problem in any comparative analysis of any taxon, our categorisation of diet represents the best current knowledge in this field and accurately captures the degree of specialisation, or lack thereof, in these taxa.

1. Drumheller, S. K., and E. W. Wilberg. 2019. A synthetic approach for assessing the interplay of form and function in the crocodyliform snout. *Zoological Journal of the Linnean Society*.

3) Minor quibbles: Your analysis includes *Pristichampsus vorax*, but *Pristichampsus* was

placed in *Boverisuchus* in 2013. In Fig. 1 you include *Pholidosaurus perbeckensis*, but there is no mention of a specimen for this species in the specimen list in the Supp Info, only reference to *Pholidosaurus* sp.

Response: Thank you for catching this error – we have changed the figure annotations to reflect the revised taxonomy of *Boverisuchus* and to correctly label the point in Fig 1 as *Pholidosaurus* sp.

For other comments, please refer to my annotations on the attached documents.

I want to like this manuscript, but I had a hard time understanding exactly how the analyses were conducted and what the results meant. I think this is mainly because of the language that is used, and the assumption that the reader is well versed in high resolution 3D geometric morphometrics and Bayesian evolutionary modelling. If you rephrase some of the text to make it more comprehensible to a broader audience, your manuscript will be much better received and the results more broadly appreciated. Once you do that, I would be happy to endorse it for publication in Proc B of the Royal Society.

Thank you for the many helpful comments, and we are confident that the changes that we have made have our manuscript more accessible for the broad readership of Proc B.

Sincerely,

Steve Salisbury

Appendix C

We thank the two reviewers for their constructive and thorough comments on this revised manuscript. Below, we detail our responses to each comment.

Referee: 1

Comments to the Author(s).

As I said in my prior review, I find this to be a very interesting and important paper. Having gone through the revised manuscript I feel that the authors have satisfactorily addressed all of my prior concerns and suggestions. I recommend acceptance once some outstanding nomenclatural issues are dealt with.

RESPONSE: Thank you to this reviewer for the kind words and the comments on nomenclature.

Given the use of the molecular topology for Crocodylia, there are some nomenclatural issues in the paper that must be addressed.

Under the molecular topology, Brevirostres is a junior synonym of Crocodylia. Authors should use Crocodylia when discussing clades based on this tree.

see lines 124, 243, 264, Figure 1 caption

RESPONSE: Thank you for pointing out the synonymy of these clades. We have replaced references to Brevirostres with "brevirostrine crocodylians," e.g., line 269-271: "Dorsoventral compression is a key aspect of variation among brevirostrine crocodylians (e.g., alligatoroids and crocodylids other than Gavialoidea) and separates brevirostrine crocodylians from other lineages (Fig. 1C)."

Line 126: "stem-crocodylian". Not sure what they mean here. The only non-crocodyloid non-alligatoroid crocodylian in the tree is Boverisuchus. What do they mean by "stem-crocodylian"?

RESPONSE: In order to avoid ambiguity with the term "stem," we have changed this section to read "Most members of Alligatoroidea and Crocodyloidea score high on PC2 and have dorsal orbits, large pterygoid flanges, and a dorsal concavity of the snout. All other clades are low on PC2, with more lateral orbits, smaller pterygoids, and more dorsoventrally compressed snouts. PC3 (5.5% of the total variance) is dominated by the dorsoventral tapering of the snout, whereas PC4 (4.2% of the total variance) reflects its lateral tapering (Fig S2)."

Line 273 and 277: Authors use "Crocodyloidea" here. Under the molecular topology Crocodyloidea includes Gavialis, so I'm not sure they actually mean Crocodyloidea. They probably mean Crocodylidae, as this would exclude Gavialis, C. affinis, and Prodiplacynodon.

RESPONSE: We have changed Crocodyloidea to Crocodylidae in these instances.

Regarding Figure 1.

First, the "A" label is missing from part A of the figure.

RESPONSE: Label A added back to figure 1

Second, the orange-ish color used for Crocodylidae in Figure 1A looks identical to the "terrestrial" color in parts B and C. This is pretty confusing as no crocodylid is terrestrial. It's not immediately apparent that the color scheme shifts between the two parts of the figure.

RESPONSE: Orange color changed to purple for better readability.

"Pristichampsus" should be "Boverisuchus".

RESPONSE: "Pristichampsus" changed to "Boverisuchus".

Pristichampsus vorax, Prodiplacynodon sp., and Crocodylus affinis are colored as "Crocodylidae" but are not members of this clade. Prodiplacynodon and Crocodylus affinis are crocodyloids but not crocodylids. Pristichampsus is not even a Crocodyloid.

RESPONSE: Clade labelled corrected

There are a number of "Extant Alligatoroidea" dots in Figure 1A that do not have species names associated with them. There is one "Extant Crocodylidae" dot next to Leidyosuchus that is also not labeled. Were these all intentional?

RESPONSE: Yes, this is intentional as this region of morphospace is to cluttered to include labels for each taxon.

Why give *Kaprosuchus* the *Mahajangasuchidae* label as opposed to *Notosuchia*? It is deeply nested in *Notosuchia* and *Kaprosuchus/Mahajangasuchidae* is never mentioned in the text?

RESPONSE: Although we do not discuss *Kaprosuchus* in the manuscript specifically, it is unique enough in its morphology and diet that we feel it may be of interest to specialist readers and thus have highlighted it in the PCA plots.

Figure 1 caption: "PC axis 2 separates the extant *Brevirostres* (alligators, caimans, and crocodiles) from stem groups (C)." This is another confusing use of *Brevirostres* and an imprecise use of "stem groups". Are the authors including *Gavialis* as a "stem group". And again, *Brevirostres* is synonymous with *Crocodylia* under the molecular topology.

Figure 3.

The "*Crocodyloidea*" black bar does not overlap with the correct taxa in the cladogram.

The "*Alligatoidea*" black bar does not overlap with the correct taxa in the cladogram.

The "*Notosuchia*" black bar does not overlap with the correct taxa in the cladogram.

RESPONSE: Position and length of clade labels corrected

Given all of these nomenclatural issues, I suggest the authors include a brief "nomenclature" section early in the paper (or less ideally, in the supplement) where they provide the names and definitions they are using. I would further recommend that if the author want to refer to the "crocodyloids and alligatoroids that aren't gavialids" they could make clear in this new Nomenclature section that they will use the term "brevirostrine crocodylians" to refer to this paraphyletic group based on molecular topologies as it distinguishes a group of crocodylians with similar snout shapes that are recovered as the clade *Brevirostres* in the morphological trees.

RESPONSE: In the interest of page limits, we have elected not to add a dedicated nomenclature section but now define brevirostrine crocodylians in-text (line 269-271)

Referee: 2

Comments to the Author(s).

I think you've done a great job addressing the issues that were raised with the initial submission. I very much appreciated the way the language of the paper has been tempered to make it more accessible to non-specialists. THANK YOU! I have the response to the reviews and re-read the manuscript in its entirety, and found the entire experience much more enjoyable than I did the first time around. I made a few minor edits (mainly grammatical/typo related) to the attached word document, and added a few comments.

RESPONSE: We thank this reviewer for their comments on this revision and we are pleased that this version is more enjoyable to them. We have updated the formatting of several citations based on Reviewer 2's attached comments.

One thing I noticed this time relates to the use of the term 'snout'. In most instances, you seem to be using it in reference to the upper jaws or more specifically the maxillary rostrum, but you don't actually specify this. Later in the text you start talking about the mandible as being separate to the snout, and then you drop in a few 'rostrums'. Given how critical the 'snout' is to this manuscript, I think it's probably worth being a bit more precise with your terminology. Rostrum (= 'the snout') includes both the maxillary rostrum and the mandibular rostrum. The maxillary rostrum is formed by the ankylosis of the right and left premaxillary, maxillary, lacrimals and prefrontal bones, and to varying degrees the right and left palatal bones and the rostral process of the frontal. The mandibular rostrum corresponds to the portion of the mandible formed by the union of the symphyseal segments of the right and left mandibular rami. Although this region of ankylosis of the mandibular rami is often called the 'mandibular symphysis', in a strict sense the symphysis refers only to the actual joint connecting the two bones.

RESPONSE: We now properly define "snout" on line 49-50: Crocodyliform skull shape is thought to vary primarily in the degree of snout (i.e., maxillary rostrum) elongation [4,5,8-12].

Once these minor edits are accepted, I'd be happy to support publication of this article in *Proc RSB*. It's great to see it reach this point. Kind regards, Steve Salisbury